medicinal chemistry/biochemistry

oxidative stress, *Coprinus comatus*, carbon tetrachloride

**Author for correspondence:**
Ana Tomas
e-mail: ana.tomas@mf.uns.ac.rs; aanaa_tomas@yahoo.com

This article has been edited by the Royal Society of Chemistry, including the commissioning, peer review process and editorial aspects up to the point of acceptance.

# Chemical composition, nutritional profile and *in vivo* antioxidant properties of the cultivated mushroom *Coprinus comatus*

Nebojša Stilinović[1], Ivan Čapo[2], Saša Vukmirović[1], Aleksandar Rašković[1], Ana Tomas[1], Mira Popović[3] and Ana Sabo[1]

[1]Faculty of Medicine, Department of Pharmacology, Toxicology and Clinical Pharmacology, and [2]Faculty of Medicine, Department of Histology and Embryology, University of Novi Sad, Hajduk Veljkova 3, 21000, Novi Sad, Serbia
[3]Faculty of Sciences, Department of Chemistry, University of Novi Sad, Trg Dositeja Obradovića 3, 21000, Novi Sad, Serbia

AT, 0000-0003-2361-872X

This study investigated the chemical and nutritional profile and antioxidative properties of cultivated *Coprinus comatus*. Proximate analysis revealed that *C. comatus* is rich in carbohydrates, dietary fibres and proteins, and could also be a valuable source of phenolics. Additionally, fat content is low, consisting mainly of polyunsaturated and omega-3 fatty acids. Furthermore, the safety profile of *C. comatus* is satisfactory, with all elements of toxicological importance within the proposed limits. Oral treatment with *C. comatus* for 42 days improved the antioxidant capabilities and ameliorated carbon tetrachloride-induced liver damage in rats, marked by decreased serum aminotransferase levels and lipid peroxidation intensity. Glutathione concentrations increased in a dose-dependent manner. Histological morphometric and immunohistochemical analysis confirmed antioxidative and hepatoprotective potential. These findings imply that cultivated *C. comatus* could be considered a nutraceutical, having beneficial nutrient and therapeutic properties.

## 1. Introduction

Significant efforts have been dedicated to finding a link between human health, nutrition and disease prevention. The concept that

diet is central to human health has raised consumer demand for dietary supplements and functional foods [1,2]. It is, therefore, not surprising that we are witnessing a growing interest in natural products with possible health benefits. For example, the National Health and Nutrition Examination Survey has shown that more than one half of the United States population uses dietary supplements regularly [2]. Similar trends can be seen in European countries [3,4].

Recently, dietary supplements or functional foods based on mushrooms have become very attractive [3,4]. Although mushrooms currently do not constitute a significant portion of the human diet, their consumption continues to increase due to their nutritional value. Apart from minerals, fibres and essential fatty and amino acids, mushrooms contain a considerable diversity of biomolecules with nutritional and medicinal properties. There are many reports identifying mushrooms as an easily accessible source of phenolics, vitamins and other natural antioxidants [5–7]. The synergy of these components is probably the leading cause of beneficial effects described in preclinical trials [8–10].

*Coprinus comatus* (O. F. Müll.) Pers. is an edible mushroom that grows wild worldwide and is very popular in Serbia [4,6,11,12] for its delicious flavour and high nutritional value. It is considered a source of valuable pharmacologically active compounds [9–11] and is thus cultivated in many countries [4,10,12]. *Coprinus comatus* owes its popularity to a myriad of bioactive properties, with its antidiabetic activity being the most widely described in the literature [9–11]. According to available literature, cultivated *C. comatus* possesses antioxidant potential similar to its wild-growing counterpart and could therefore represent an excellent dietary option for fighting free radicals. These studies have dealt mainly with chemical characterization and the *in vitro* antioxidant properties of *C. comatus*. However, none have investigated the antioxidant properties of *C. comatus* using an *in vivo* model [4,12,13], apart from a single study by our research group that explored the antioxidant activity of short-term treatment with *C. comatus* in rats [14]. Therefore, besides chemical characterization, the current study aimed to evaluate the *in vivo* antioxidant effects of long-term administration of cultivated *C. comatus* mushroom in a rat model treated with carbon tetrachloride ($CCl_4$) as a pro-oxidant substance.

# 2. Materials and methods

## 2.1. Mushroom samples

A commercial preparation of *C. comatus* (O. F. Müll.) Pers. was used for the experiments. This is prepared from whole, cultivated, lyophilized, *C. comatus* mushrooms (100%) and sold as a fine powder (20 mesh). It is available in Serbian pharmacies as a food supplement. Market release of this preparation was authorized by the Ministry of Health of the Republic of Serbia. Mushroom preparation samples used for the experiments were stored in desiccators and protected from light.

## 2.2. Chemicals

Ethyl acetate, acetonitrile and hexane were of HPLC grade from Fisher Scientific (Pittsburgh, PA, USA). The fatty acids methyl ester (FAME) reference standard mixture 37 (standard 47885-U) was purchased from Sigma-Aldrich Chemical Co. (St Louis, MO, USA). Water was treated in a Millipore Elix Essential 10 UV water purification system (Billerica, MA, USA). Folin–Ciocalteu's reagent, methanol, ethanol, dimethylsulfoxide (DMSO), aluminium chloride, p.a. formic acid, acetic acid and trichloroacetic acid (TCA) were obtained from E. Merck (Darmstadt, Germany). Concentrated 69% nitric acid (ccHNO$_3$) ('for trace elements analysis' grade) and 30% hydrogen peroxide ($H_2O_2$) were purchased from J. T. Baker Chemical Co. (Phillipsburg, NJ, USA). All the plastic and glassware were cleaned by soaking in a 20% hydrochloric solution overnight and then in 20% nitric acid overnight and finally rinsed with Millipore water. The Pb, Cd, Hg, As, Fe, Mg, Cu, Zn and Se stock standard solutions (1000 µg ml$^{-1}$) were supplied by J. T. Baker Chemical Co. (Phillipsburg, NJ, USA). Reference standards of the phenolic compounds, including gallic acid and quercetin, were obtained from Sigma-Aldrich (St Louis, MO, USA) or Fluka Chemie GmbH (Buchs, Switzerland). Thiobarbituric acid, glutathione, glutathione reductase from baker's yeast, EDTA, NADPH tetrasodium salt, 5,5′-dithiobis (2-nitrobenzoic acid), $CCl_4$ and urethane were purchased from Sigma-Aldrich Chemical Co. (St Louis, MO, USA) or Carl Roth GmbH (Karlsruhe, Germany). Total protein assay was purchased from Sentinel Diagnostics (Milan, Italy). All the other chemicals were of analytical grade and obtained from conventional sources.

## 2.3. Nutrient and non-nutrient composition of Coprinus comatus

### 2.3.1. Proximate analysis

Moisture, ash, proteins, fat, carbohydrates and dietary fibre in the mushroom sample were determined using accepted analysis methods of the Association of Official Analytical Chemists (AOAC) [15]. Moisture content was determined by drying the sample at 105°C according to the AOAC method 925.10. Ash content was determined by incineration at 600°C according to the AOAC method 942.05. Total nitrogen (N) was determined by the Kjeldahl method, AOAC method 950.36, and protein was calculated as total N × 6.25. Fat content was determined using the Soxhlet apparatus with hexane, following AOAC 963.15 method. The carbohydrate content was calculated by difference: 100 − (fat + protein + crude fibre). Crude fibre content was determined using a fibre digester according to the AOAC method 973.18. Energy was calculated according to the following equation: energy (kcal) = 4 × (protein (g) + carbohydrate (g)) + 9 × (fat (g)).

### 2.3.2. Fatty acids determination

Fatty acids were determined after the extraction and transesterification procedures described by Pojić et al. [16]. Briefly, lipids were extracted with chloroform/methanol mixture (2 : 1 v/v), and the extracts obtained were dried by vacuum evaporation at 40°C. Further, methyl esters were prepared from the extracted lipids by transesterification with 14% boron trifluoride in methanol. The fatty acid profile was analysed with an Agilent 7890A gas chromatograph system (Santa Clara CA, United States) with a flame ionization detector equipped with a fused silica capillary column (DB-WAX 30 m, 0.25 mm, 0.50 µm). The carrier gas was helium with a flow rate of 1.26 ml min$^{-1}$. The fatty acid peaks were identified by comparing relative retention times of FAME from the sample with retention times of standards from Sigma-Aldrich Chemical Co. 37 component FAME mix (St Louis, MO, USA) and with data from an internal data library. The results were expressed as a weight percentage of crude fat.

### 2.3.3. Mineral and trace elements and toxic heavy metals analysis

Coprinus comatus dry powder preparation samples were digested using concentrated $HNO_3$ and $H_2O_2$ and Ethos One, Milestone microwave (Sorisole, Italy) with an internal temperature sensor. The operational conditions and the heating programme were carried out according to the manufacturers' recommendations. Digests were diluted with ultrapure deionized water to 25 ml in a glass flask and transferred to a previously acid-cleaned polypropylene vessel for further analysis. A Varian SpectrAA 10 (Varian Techtron Pty Limited, Mulgrave, Australia) model atomic absorption spectrometer (AAS) with deuterium background correction was used for determination of mineral element (Mg), trace element (Fe, Cu, Zn, Se) and toxic heavy metal (Pb, Cd, Hg, As) contents. The assembly was operated from an interfaced computer running SpectrAA software. The samples were analysed according to the procedures and conditions described in our previous publication [4]. Coprinus comatus samples were analysed in triplicate ($n = 3$), and blank samples were included to control for possible contamination.

### 2.3.4. Phenolic profile

#### 2.3.4.1. Extraction procedure

The commercial preparation of C. comatus mushroom samples was extracted with pure methanol or 70% (v/v) methanol. Namely, the samples (1 g) were extracted by maceration with 50 ml of solvent during 24 h at room temperature with constant shaking. After filtration, the maceration process was repeated with 50 ml of solvent for 3 h with shaking. Then, the solvent was evaporated in vacuo at 40°C, and the obtained dried extracts were redissolved in DMSO to obtain stock solutions for further analysis.

#### 2.3.4.2. Determination of total phenolics

Phenolic content of mushroom extracts was assayed using the method given in the literature [17]. One hundred microlitres of 0.5% mushroom extracts were mixed with 500 µl of Folin–Ciocalteu reagent and 400 µl of 7.5% sodium carbonate solution in test tubes. After being vortexed and incubated in the dark for 2 h, absorbance was measured at 740 nm. All samples were made in triplicate. The concentrations of phenolic compounds were calculated from the standard gallic acid graph and expressed as mg of gallic acid equivalents (GAE) per g of dry extract.

### 2.3.4.3. Determination of total flavonoids

The total content of flavonoids was determined according to the aluminium chloride colorimetric method [18], adapted for microtitre plates. Namely, 30 μl of extract or standard solution was diluted with 90 μl of methanol and 6 μl of 10% aluminium chloride, 6 μl of 1 mol l$^{-1}$ potassium acetate and 170 μl of distilled water was added. Absorbance was determined after 30 min at 415 nm. All samples were made in triplicate, and mean values of flavonoid content were expressed as mg of quercetin equivalents (QE) per g dry weight of extract and calculated according to the standard calibration curve.

### 2.3.4.4. LC–MS/MS analysis

LC–MS/MS analysis of the selected phenolics was done after Orčić et al. [19]. Extracts were diluted with a mixture of mobile phase solvents A (0.05% aqueous formic acid) and B (methanol), in a 1 : 1 ratio, to obtain a final concentration of 2 mg ml$^{-1}$. Working standards, ranging from 1.53 ng ml$^{-1}$ to 25.0 × 10$^3$ ng ml$^{-1}$, were prepared by serial 1 : 1 dilutions of the standard mixture with solvent (A : B = 1 : 1). Samples were analysed with Agilent Technologies 1200 Series HPLC coupled with Agilent Technologies 6410A Triple Quad tandem mass spectrometer with electrospray ion source (Santa Clara, CA, USA), controlled by Agilent Technologies MassHunter Workstation software—Data Acquisition (v. B.03.01). The injection volume was 5 μl. Compounds were separated with Zorbax Eclipse XDB-C18 (50 × 4.6 mm, 1.8 μm) rapid resolution column heated at 50°C. The mobile phase was delivered at a flow rate of 1 ml min$^{-1}$ in a gradient mode (0 min 30% B, 6 min 70% B, 9 min 100% B, 12 min 100% B, re-equilibration time 3 min). Eluted components were detected by MS, using the ion source parameters as follows: nebulization gas (N$_2$) pressure 40 psi, drying gas (N$_2$) flow 9 l min$^{-1}$ and temperature 350°C, capillary voltage 4 kV and negative polarity. Data were acquired in dynamic MRM mode, using the optimized compound-specific parameters. For all the compounds, peak areas were determined using Agilent MassHunter Workstation Software—Qualitative Analysis (v. B.03.01). Calibration curves were plotted, and samples' concentrations calculated using the OriginLabs Origin Pro (v. 8.0) software. All samples were made in triplicate, and the results were expressed in mg per g of the dry mushroom sample weight.

## 2.4. Effects of *Coprinus comatus* against carbon tetrachloride-induced liver injury in rats

### 2.4.1. Laboratory animals

White Wistar laboratory rats of both sexes (three male and two female rats in separate cages for each group), weighing 250–300 g and ages up to three months, were obtained from local animal facilities (Military Medical Academy, Belgrade, Serbia). Animals were housed in the UniProtect airflow cabinet (Ehret GmbH, Emmendingen, Germany) and standard Plexiglas cages at a constant 22 ± 1°C room temperature, 55% ± 1.5% humidity and with standard circadian rhythm (12 h day/night cycle). They were allowed free access to tap water and standard pelleted laboratory rodent feed (Veterinary Institute Subotica, Serbia) during the whole experiment. Weights of all rats were measured at the beginning and at the end of the experiment. Health status of the animals was checked every day during the experiment. There were no health issues in any of the experimental groups prior to CCl$_4$ administration. The experimental procedures were conducted per the European Directive (2010/63/EU) for animal experiments, and they were reviewed and approved by the Ethics Committee for Protection and Welfare of Experimental Animals at the University of Novi Sad, Serbia (Approval No. III-2011-07).

### 2.4.2. Experimental procedures

The antioxidant and hepatoprotective properties of a commercial preparation of *C. comatus* were investigated using a rat model of CCl$_4$-induced hepatotoxicity. After a week of adaptation before the experiments, 30 rats were randomly divided into six groups ($n = 5$) as follows:

Sal group—rats received saline per os (p.o.) (2 ml kg$^{-1}$ day$^{-1}$) for 42 days and olive oil intraperitoneally (i.p.) (2 ml kg$^{-1}$) on the last day.

SalCCl$_4$ group—rats received saline p.o. (2 ml kg$^{-1}$ day$^{-1}$) for 42 days and CCl$_4$ i.p. (2 ml kg$^{-1}$) on the last day.

C1CCl$_4$ group—rats received *C. comatus* p.o. (0.835 g kg$^{-1}$ day$^{-1}$) for 42 days and CCl$_4$ i.p. (2 ml kg$^{-1}$) on the last day.

C2CCl$_4$ group—rats received *C. comatus* p.o. (1.67 g kg$^{-1}$ day$^{-1}$) for 42 days and CCl$_4$ i.p. (2 ml kg$^{-1}$) on the last day.

C3CCl4 group—rats received *C. comatus* p.o. (3.34 g kg$^{-1}$ day$^{-1}$) for 42 days and CCl4 i.p. (2 ml kg$^{-1}$) on the last day.

C3 group—rats received *C. comatus* p.o. (3.34 g kg$^{-1}$ day$^{-1}$) for 42 days and olive oil i.p. (2 ml kg$^{-1}$) on the last day.

*Coprinus comatus* treatment groups were administered an aqueous suspension of mushroom orally in three different doses using an intragastric probe for rats. An administered dose of 3.34 g kg$^{-1}$ correlates approximately to one standard serving of 30 g dry weight mushroom for the average person (60 kg), calculated using the Food and Drug Administration's Human Equivalent Dose formula [20]. The other two doses of mushroom were half and quarter of the previously mentioned dose. Saline groups received an orally administered 2 ml kg$^{-1}$ dose of isotonic saline using an intragastric probe for rats. Two hours after the final administration, groups 1 and 6 were treated with olive oil, i.p. Groups 2–5 were treated i.p. with 50% CCl4 (olive oil : CCl4 = 1 : 1) at a dose of 2 ml kg$^{-1}$. All rats were euthanized 24 h after the administration of CCl4 or olive oil. Animals were under urethane anaesthesia (5 ml kg$^{-1}$ i.p. of 25% solution) and exsanguinated with intracardial punction. The blood was collected and centrifuged for 10 min at 3000 r.p.m. and 4°C for the separation of serum. The liver was also rapidly collected, rinsed with ice-cold saline, measured and dissected. One segment of the liver sample was quickly frozen with liquid nitrogen and stored in Arctiko ULUF 550 freezer at −80°C (Esbjerg, Denmark) for further analysis, and the rest was fixed in 10% buffered formalin for histopathological examination. A graphical overview of the study design is available in the electronic supplementary material, S1.

### 2.4.3. Biochemical parameters analysis

#### 2.4.3.1. Aspartate and alanine aminotransferase assays

The serum was used for the analysis of the enzymatic activity of aspartate (AST) and alanine (ALT), markers of liver damage. The serum activities of AST and ALT were measured using clinical biochemistry, fully automated analyser Olympus AU400 (Hamburg, Germany).

#### 2.4.3.2. Lipid peroxidation, reduced glutathione, glutathione peroxidase and catalase assays

Homogenate was prepared from 1 g of liver tissue, which was minced, homogenized in a Tris-buffer solution (pH 7.4; organ : buffer = 1 : 3) and divided into two portions. One was used for lipid peroxidation intensity determination, and the other was centrifuged on Mikro 22R for 20 min at 10 000 r.p.m. and 4°C (Andreas Hettich GmbH, Tuttlingen, Germany). The obtained supernatant was used for the assays of total protein concentration, reduced glutathione (GSH), glutathione peroxidase (GPx) and catalase (CAT). Lipid peroxidation (LPx) intensity was measured through the malondialdehyde (MDA) concentrations obtained with thiobarbituric acid reactive substances (TBARS) method [21]. Total protein concentration was measured using a Sentinel Diagnostic commercial assay kit (Milan, Italy) according to the manufacturer's protocol. GSH concentration was determined after the protocol described in Kapetanović and Mieyal [22], GPx activity after Chiu *et al.* [23] and CAT activity after Aebi [24]. All measurements were obtained on an Agilent 8453 UV/Vis spectrophotometer (Santa Clara, CA, United States).

### 2.4.4. Histopathological analysis

#### 2.4.4.1. Tissue preparation and tissue microarray formation

The *ca* 5 × 5 mm tissue biopsy samples of the right medial liver lobe of each animal were fixed in 10% neutral-buffered formalin solution for 24 h at 4°C. After fixation, liver tissue was dehydrated in isopropyl alcohol and embedded in Histowax paraffin (Duiven, The Netherlands). Using a tissue punch, two cylindrical samples, 2 mm in diameter, were taken from each paraffin tissue block and transferred to a paraffin block with 10 holes (2 holes per animal). In this way, we formed six 2 mm core tissue microarray (TMA) blocks with 10 tissue samples representing one experimental group (five animals analysed per group). Each of the TMA blocks was cut on the Leica rotary microtome (Wetzlar, Germany) at 5 µm.

#### 2.4.4.2. Haematoxylin-eosin and immunohistochemical staining

For each liver TMA block, the first slide section was stained using the standard haematoxylin-eosin (H&E) method, and the next four serial sections were immunohistochemically stained. Immunohistochemical staining included the following primary antibodies: Cusabio rabbit anti-human transcription factor

cyclooxygenase-2 (COX-2) in a 1 : 100 dilution (College Park, MD, USA); Cusabio rabbit anti-human superoxide dismutase 2 (SOD2) in a 1 : 100 dilution (College Park, MD, USA); Cusabio rabbit anti-human cytochrome P450 2E1 protein (CYP2E1) in a 1 : 100 dilution (College Park, MD, USA) and Lab Vision mouse anti-cytochrome c (CYTC) Ab-2 in a 1 : 50 dilution (Thermo Fisher Scientific, Waltham, MA, USA); and used the UltraVision LP Detection System using HRP Polymer & DAB Chromogen (Thermo Fisher Scientific, Waltham, MA, USA). All the antibodies were applied for 30 min at room temperature following antigen retrieval using citrate buffer (pH 6.0) in a microwave oven at 850 W for 20 min. Mayer's haematoxylin was used as a counterstain for immunohistochemistry prior to mounting and coverslipping (Bio-Optica, Milan, Italy) the slides. Prepared slides were viewed using the Leica DMLB microscope (Wetzlar, Germany) and each of 10 tissue sections (2 × 2 mm in diameter) from each TMA block were separately photographed on the Leica MC 190 HD camera (Wetzlar, Germany) using Live Image Builder software from the Leica Application Suite (Wetzlar, Germany).

### 2.4.4.3. Histomorphometric analysis

To quantify the degree of liver tissue damage, the areal density of necrosis, hepatocellular ballooning, micro/macrovesicular steatosis, apoptotic hepatocytes, inflammatory cells (activity) and preserved hepatocytes were measured. The areal density was obtained by counting points over the mentioned structures and dividing with a total number of points from each analysed H&E-stained liver tissue section photograph (2 × 2 mm). The analysis was performed using ImageJ 1.45 image measurement software with the plug-in for the Windows platform, as described by Schneider *et al.* [25].

### 2.4.5. Statistical analysis

Statistical analysis was performed using IBM SPSS statistical software, version 19.0 (IBM Corp., Armonk, NY, USA). Data were reported as the mean ± standard deviation (s.d.) or standard error of the mean (s.e.m.). One-way analysis of variance (ANOVA) or Kruskal–Wallis test was employed for the comparisons between experimental groups. *Post hoc* testing for ANOVA was performed using Tukey's test. Mann–Whitney U test was used for *post hoc* testing in Kruskal–Wallis analysis. The difference between groups was considered statistically significant for a *p*-value less than 0.05 ($p < 0.05$).

# 3. Results and discussion

## 3.1. Nutrient and non-nutrient composition of *Coprinus comatus*

### 3.1.1. Proximate analysis

Results of the proximate analysis of *C. comatus* commercial preparation are presented in table 1. Only a few studies have determined the composition of nutrients in *C. comatus*, either wild-growing or cultivated [12,26]. This commercial preparation represents dry powder of cultivated *C. comatus*, and results of the proximate analysis are comparable with those obtained by other authors [12,26], despite slight differences in methodology. Similar values were obtained for fat content, but there are differences in ash, protein, carbohydrate and energy values compared with the results of Stojković *et al.* [12]. The total energy calculation determined a lower value in our study, largely due to higher ash content and the exclusion of crude fibre. Using the same methodology for protein analysis, we found higher protein content in *C. comatus* than Stojković *et al.* [12].

Composition of nutrients in *C. comatus* from our study were similar to those reported by Tsai *et al.* [26], with the significant exceptions of ash (13.24 : 8.44) and crude fibre content (21.13 : 12.48). Elsewhere [7], *C. comatus* was one of four wild-growing mushrooms investigated. Despite other disagreements, no significant difference in ash content was found between wild *C. comatus* from Portugal and our commercial preparation [7].

A study by Yang *et al.* [27], which included six commercially grown mushrooms, but not *C. comatus*, showed that these mushrooms consisted of 39.6%–63.9% carbohydrates and 15.4%–26.7% proteins. These results are similar to those obtained for our commercial mushroom. On the other hand, the fat content of our *C. comatus* was lower than the lowest value for their mushrooms (2.16%–9.23%), and crude fibre content was considerably higher than their 16.96%. These findings support our opinion that *C. comatus* can provide superior nutritional benefit to consumers due to their low fat and high protein and fibre content. Especially interesting is the high fibre content, because mushrooms are

**Table 1.** Proximate composition of a lyophilized commercial preparation of cultivated *C. comatus* mushroom (mean ± s.d.; *n* = 3).

| parameter | concentration |
|---|---|
| moisture (g/100 g) | 8.63 ± 0.24 |
| dry matter (g/100 g) | 91.37 ± 0.65 |
| ash (g/100 g d.m.)[a] | 13.24 ± 0.19 |
| proteins (g/100 g d.m.) | 23.07 ± 0.28 |
| fat (g/100 g d.m.) | 2.04 ± 0.03 |
| carbohydrates (g/100 g d.m.) | 40.42 ± 0.48 |
| dietary fibre (g/100 g d.m.) | 21.13 ± 0.34 |
| energy (kcal/100 g) | 272.32 ± 0.84 |
| C16 : 0[b] | 12.93 ± 0.18 |
| C18 : 0 | 0.97 ± 0.02 |
| C18 : 1n7 | 1.05 ± 0.07 |
| C18 : 1n9 | 5.82 ± 0.11 |
| C18 : 2n6 | 73.39 ± 0.24 |
| C18 : 3n3 | 1.06 ± 0.02 |
| C20 : 3n3 | 0.65 ± 0.02 |
| C20 : 5n3 | 0.91 ± 0.03 |
| SFA[c] | 15.84 ± 0.08 |
| MUFA | 7.42 ± 0.07 |
| PUFA | 76.73 ± 0.30 |

[a]d.m., dry matter.
[b]Main fatty acids: C16 : 0 (palmitic acid), C18 : 0 (stearic acid), C18 : 1n7 (cis-11-octadecenoic acid), C18 : 1n9 (oleic acid), C18 : 2n6 (linoleic acid), C18 : 3n3 (α-linolenic acid), C20 : 3n3 (eicosatrienoic acid), C20 : 5n3 (eicosapentaenoic acid); 11 more fatty acids were identified in relative percentage lower than 0.5. Concentration of fatty acids is expressed in relative percentage.
[c]SFA, saturated fatty acids; MUFA, monounsaturated fatty acids; PUFA, polyunsaturated fatty acids.

underutilized as a source of dietary fibres, and their cell walls contain non-digestible carbohydrates, such as chitin, β-glucans and mannans [28].

The fatty acid composition results obtained for the commercial preparation of *C. comatus* are also presented in table 1. A total of 19 fatty acids were identified, but only 8 had a relative concentration of more than 0.5. The most prominent fatty acid found was linoleic acid, which contributed to the PUFAs (polyunsaturated fatty acids) being the most widely represented fatty acid group. In the second place was palmitic acid, as the significant SFA (saturated fatty acid), and the most abundant MUFA (monounsaturated fatty acid) was oleic acid. It is essential to point out that three ω-3 fatty acids were also identified with relative concentration more than 0.5: α-linolenic acid, eicosatrienoic acid and eicosapentaenoic acid.

The fatty acid composition of our *C. comatus* preparation was almost analogous with that of its wild-growing, same species counterpart from Portugal [5]. Linoleic, palmitic and oleic acids were also described by Stojković *et al.* [12] as the primary fatty acids in their samples of wild-growing and cultivated *C. comatus*. Linoleic acid had the highest relative concentration in a sample of *C. comatus* from Turkey as well, but in this case, the fatty acid profile showed that palmitoleic and stearic acids had a significant share, which was not the case in our present study [29].

When comparing the fatty acid composition of the studied *C. comatus* with other commercial edible mushrooms such as button, oyster or shiitake, it becomes evident that linoleic, palmitic and oleic acids are the primary fatty acids common to all of them [6]. Our results are consistent with the observation that linoleic acid is the primary fatty acid in almost every wild-growing and cultivated edible mushroom [29,30]. This could be explained by the fact that linoleic acid is the precursor of 1-octen-3-ol, which is one of the main aromatic compounds in most fungi [30]. Taking this information together, unsaturated fatty acids predominate over saturated fatty acids in mushrooms, which contributes to their nutritional food value [5,30].

**Table 2.** Concentration of the investigated elements (mean ± s.d.) in *C. comatus* mushroom, calculation of the element intake through consumption of one serving (300 g fresh weight or 30 g dry weight) of mushroom (mean ± s.d.; *n* = 3) and percentage coverage of recommended or tolerable upper level of metal intake (in brackets).

| element | concentration (mg kg$^{-1}$ d.w.)[a] | intake per one serving (mg) |
|---|---|---|
| Pb | 0.172 ± 0.04 | 0.005 ± 0.001 (0.33 of PTWI)[b] |
| Cd | 0.14 ± 0.08 | 0.004 ± 0.003 (1.0 of PTWI) |
| Hg | 0.019 ± 0.00 | 0.001 ± 0.000 (0.19 of PTWI) |
| As | 0.38 ± 0.03 | 0.011 ± 0.001 (1.27 of PTWI) |
| Fe | 1471 ± 51.32 | 44.13 ± 1.540 (294.2 of RDI)[c] |
| Mg | 1334 ± 47.69 | 40.02 ± 1.431 (10.0 of RDI) |
| Cu | 10.17 ± 0.63 | 0.305 ± 0.010 (33.9 of RDI) |
| Zn | 31.73 ± 1.92 | 0.952 ± 0.058 (10.0 of RDI) |
| Se | 0.51 ± 0.08 | 0.015 ± 0.003 (30.6 of RDI) |

[a]d.w., dry weight.
[b]PTWI, provisional tolerable weekly intake.
[c]RDI, recommended daily intake.

### 3.1.2. Analysis of elements

All investigated element concentrations are expressed on a dry weight basis. For the calculations of the element intake through consumption of commercial preparation of cultivated *C. comatus*, we took 60 kg as the weight of an average consumer. Furthermore, we assumed that a serving of 30 g dry matter is equal to 300 g of fresh weight mushroom, as described in our previous publication [4]. The results of element concentrations, element intake and per cent coverage of provisional tolerable weekly intakes (PTWI) or recommended daily intakes (RDI) are presented in table 2.

According to the literature, two main factors affecting element content in the mushroom are species dependence and substrate composition. Wild-growing *C. comatus* has been found to have a high capacity for the accumulation of toxic elements from its substrate or the atmosphere [31,32]. Some investigators took advantage of metal bioaccumulation and enriched *C. comatus* with trace elements such as selenium, to achieve better pharmacodynamic properties [10]. In the present study, none of the toxic metal concentrations (Pb, Cd, Hg and As) nor PTWI exceeded the statutory limits set by FAO/WHO [33,34]. These results are consistent with our previous findings [4]. Toxic element concentrations in our *C. comatus* are much lower than in wild-growing *C. comatus* or many other edible mushrooms from different locations, leading us to the conclusion that cultivated mushrooms could be safer in that respect [32,35].

Besides elements of toxicological importance, we have investigated elements with a positive impact on health. Mushrooms can accumulate large amounts of essential macro- and micro-elements in their fruiting bodies [35,36]. Among them, iron was present in the highest concentration in the present sample (1471 ± 51.32 mg kg$^{-1}$ d.w.), which is almost three times greater than the daily requirement for adult men (15 mg day$^{-1}$) [37]. In the study of 15 different mushroom species by Yamac *et al.* [32], the range of Fe concentrations was between 110 and 11 460 mg kg$^{-1}$, the second-highest being determined in the *C. comatus* sample (3640 mg kg$^{-1}$ d.w.). Our previous investigation of cultivated mushrooms revealed similar results, with wide variations in iron content between different species [4]. In the current study, our sample of *C. comatus* also had a high concentration of magnesium (1334 ± 47.69 mg kg$^{-1}$ d.w.), and one standard serving of this mushroom can provide 10% of Mg RDI [37]. This concentration of Mg is within the values listed in review articles for wild-growing and cultivated mushrooms [35,36]. Copper levels (10.17 ± 0.63 mg kg$^{-1}$ d.w.) were found to be a little below values (10.60 mg kg$^{-1}$ d.w.) for most edible mushrooms described in the literature [32,35–36]; however, consumption of 30 g of cultivated *C. comatus* still ensures one-third of the RDI [37].

Interestingly, copper has antioxidant and pro-oxidant properties. It is required for the function of SOD, a key antioxidant enzyme, but when it is present in excess, it promotes the formation of reactive oxygen species. Zinc is also required for the function of SOD and many other enzymes, and as an essential trace element, it is second in abundance of all biometals in the human body. According to European Food Safety Authority (EFSA) recommendations, consumption of *C. Comatus*, studied here, can provide 10% of zinc RDI [37]. Along with copper and zinc, selenium is the third essential element for antioxidant

**Table 3.** Concentration of phenolics in *C. comatus* mushroom expressed on a dry weight basis (mean ± s.d.; *n* = 3).

| component | *C. comatus* (pure methanol) | *C. comatus* (70% methanol) |
|---|---|---|
| *p*-Hydroxybenzoic acid (µg g$^{-1}$) | 11.73 ± 0.58 | 11.41 ± 1.17 |
| Protocatechuic acid (µg g$^{-1}$) | 0.11 ± 0.02 | 0.13 ± 0.03 |
| Cinnamic acid (µg g$^{-1}$) | 4.07 ± 0.66 | 4.34 ± 0.27 |
| *p*-Coumaric acid (µg g$^{-1}$) | 8.86 ± 0.71 | 10.48 ± 0.94 |
| Caffeic acid (µg g$^{-1}$) | 0.15 ± 0.03 | 0.15 ± 0.02 |
| Quinic acid (µg g$^{-1}$) | 1.59 ± 0.43 | 9.10 ± 1.39[a] |
| total (µg g$^{-1}$)[b] | 26.52 ± 2.23 | 35.62 ± 3.19[a] |
| total phenolic content (mg GAE/g d.w. extract)[c] | 102.25 ± 3.38 | 107.02 ± 2.42 |
| total flavonoid content (mg QE/g d.w. extract)[d] | 0.59 ± 0.09 | 0.39 ± 0.08[a] |

[a]$p < 0.05$ versus pure methanolic extract.
[b]Sum of the phenolic compounds concentrations obtained by LC-MS analysis.
[c]Concentration obtained by Folin–Ciocalteu assay (GAE, gallic acid equivalent).
[d]Concentration obtained by flavonoid-aluminium chloride complex assay (QE, quercetine equivalent).

enzymes of mammals. It is a component of various forms of glutathione peroxidase enzyme [10]. Results for selenium presented in table 2 indicate that *C. comatus* can be a good source of this trace element.

### 3.1.3. Phenolic profile

Traditionally, plants were considered to be the primary source of phenolic compounds [4]. However, it has been found that mushrooms also have a considerable quantity of phenolics, and numerous reports reveal the high antioxidant capacity of cultivated and wild-growing mushrooms [4–6,38]. Antioxidant capacity correlates closely with total phenolic content, which indicates that phenolics could be the most prominent contributors to the antioxidant activity of edible mushrooms [4,7,35]. Therefore, the Folin–Ciocalteu method was used to estimate the total phenolic content of mushroom extracts. It must be noted that this reagent does not react exclusively with phenolics and has a tendency to overestimate their quantity. Nevertheless, this method was widely employed prior to quantification of phenolics using liquid chromatography [4,12,39].

We used two assays for spectrometric analysis of phenolic compounds, Folin–Ciocalteu assay and flavonoid-aluminium chloride complex assay (table 3). Differences between methanolic extracts are evident in LC-MS analysis of phenolic compounds and total flavonoid content. Pure methanol extraction gave better results for total flavonoid content (0.59 ± 0.09 versus 0.39 ± 0.08), but 70% methanolic extract showed a higher concentration of phenolics in LC-MS analysis (35.62 ± 3.19 versus 26.52 ± 2.23). Phenolics can be subdivided into several groups: flavonoids with subgroups, phenolic acids with subgroups, stilbenes, lignans and other [40]. It seems that phenolic acids are the most common phenolics found in mushrooms, which is consistent with our findings [7,12,39]. In both extracts, it was possible to quantify six different phenolic acids, namely *p*-hydroxybenzoic, protocatechuic, cinnamic, *p*-coumaric, caffeic and quinic acids (figure 1). The main difference between extracts for LC-MS analysis was in quinic acid concentration (9.11 ± 1.39 for 70% methanol versus 1.59 ± 0.43 for pure methanol). Quinic acid belongs to the cyclohexane carboxylic group of phenolic acids, and according to results of Tešanović *et al.* [40], it seems that water extraction is suitable for this type of phenolic acids. So, diluting methanol with water provided a higher concentration of quinic acid in the extract of our *C. comatus*, though not as high as in the previously mentioned study of Tešanović *et al.* [40]. With respect to flavonoids, there are contradictory data in the literature. A recent systematic review showed that edible mushrooms could not synthesize flavonoids, nor absorb them from enriched substrates, which makes spectrometric methods for flavonoid content analysis useless in their opinion [41]. On the contrary, there are reports which display various flavonoid group phenolics isolated from mushrooms [41,42]. However, further investigations are needed to clarify these results, because the concentration of phenolics can be strain-specific, but mainly it depends on factors such as cultivation techniques, growing conditions, processing and storage conditions, as well as solvent extraction procedures [39,40].

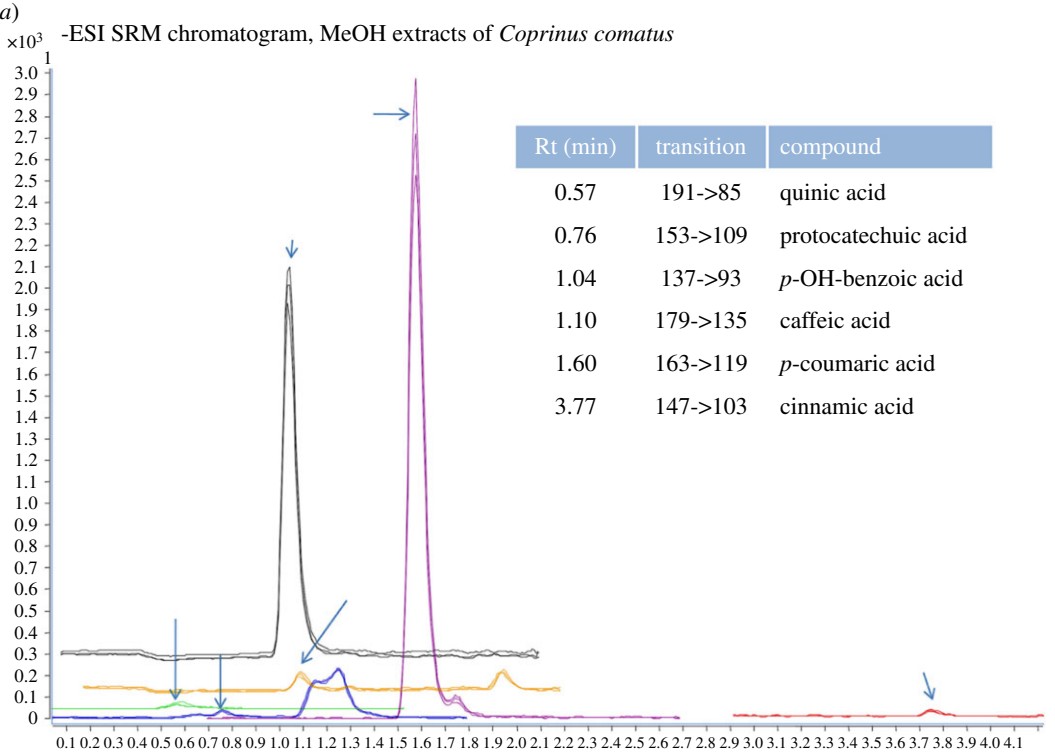

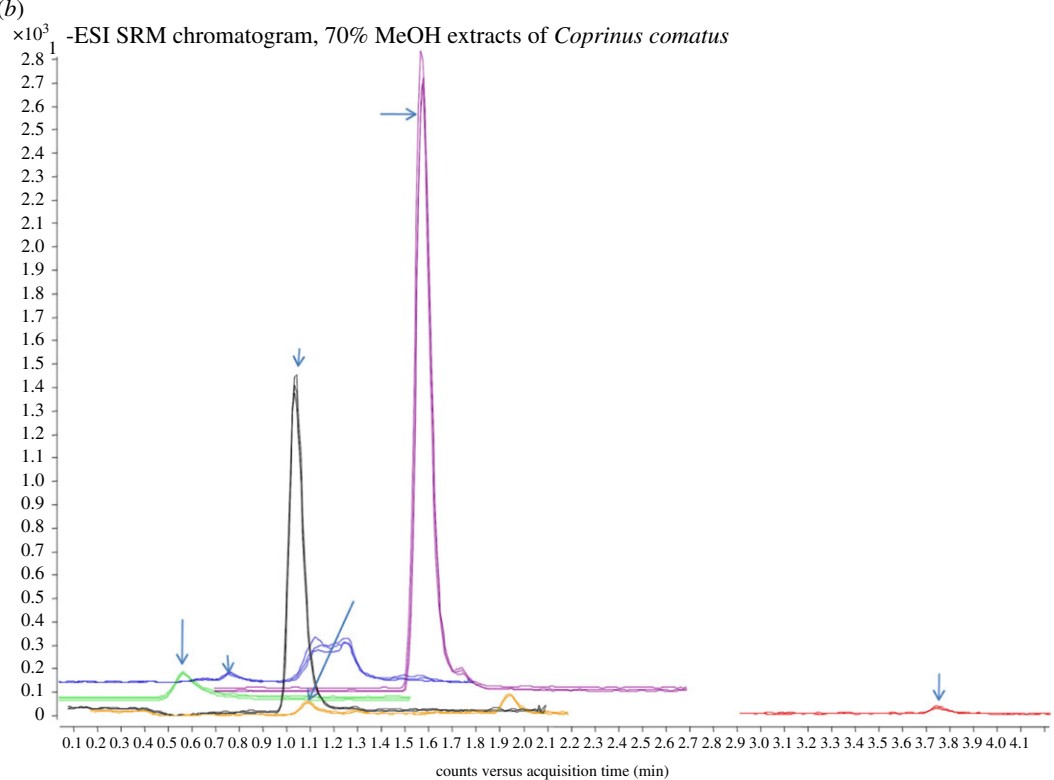

**Figure 1.** LC-MS chromatograms of the selected phenolics in methanolic extracts of *C. comatus*. Rt — retention time; MeOH — methanol.

## 3.2. Antioxidant and hepatoprotective effects of *Coprinus comatus*

### 3.2.1. Biochemistry assays

$CCl_4$-induced hepatotoxicity is one of the most frequently used models for the investigation of *in vivo* hepatoprotective properties. The mechanism of $CCl_4$'s hepatotoxicity is in its metabolism by cytochrome

**Table 4.** Weight of rats at the beginning and at the end of the experiment and weight change (mean ± s.e.m.).

| groups | start weight (g) | end weight (g) | weight change |
|---|---|---|---|
| Sal | 267.20 ± 8.78 | 364.20 ± 15.41 | 97.00 ± 13.19 |
| SalCCl$_4$ | 272.50 ± 11.15 | 274.50 ± 7.96[a] | 2.00 ± 4.02[a] |
| C1CCl$_4$ | 290.00 ± 6.56 | 362.75 ± 8.41 | 72.75 ± 10.29 |
| C2CCl$_4$ | 271.75 ± 11.67 | 340.75 ± 23.96 | 69.00 ± 13.03 |
| C3CCl$_4$ | 263.60 ± 9.90 | 335.00 ± 19.01 | 71.40 ± 10.41 |
| C3 | 277.83 ± 8.29 | 345.50 ± 16.27 | 67.67 ± 5.35 |

[a]$p < 0.05$ versus all experimental groups.

P450 to a trichloromethyl (·CCl$_3$) and trichloromethyl peroxy (ClCOO·) radical. These free radicals lead to a sequence of events with lipid peroxidation as a critical process of liver cell death [8,14,43].

The first *in vivo* parameter obtained was the weight of animals. There was no statistically significant difference between the weights of animals at the beginning of the experiment. A statistically significant difference between the groups was reached for weight and weight change at the end of the experiment (table 4). The highest weight gain was observed in the group treated with saline only (Sal), while the lowest was in the SalCCl$_4$ group. According to results from table 4, it seems that pretreatment with *C. comatus* prevented the weight loss effect of CCl$_4$. A similar influence on weight could be found in studies employing the same CCl$_4$ model of hepatotoxicity but using different hepatoprotective substances [44,45]. Furthermore, one report demonstrated that CCl$_4$ treatment alone could provoke more than 15% loss of total body weight of rats in only 48 h [46].

Biochemical analysis included measuring the serum concentration of ALT, AST and the level of several oxidative stress parameters (table 5). This investigation is a logical extension of previously published data concerning the *in vitro* antioxidant activity of cultivated *C. comatus* [4].

Damage of the hepatocyte membrane leads to leakage of aminotransferases. Thus, the serum concentration of ALT and AST relates positively to the degree of liver injury. Compared with the 42-day saline treatment, the 3.34 g kg$^{-1}$ water suspension dose of *C. comatus* did not affect the serum level of AST, but did change ALT concentration. Specifically, animals from the C3CCl$_4$ group had significantly lower concentrations of ALT than animals receiving CCl$_4$ that were pretreated only with saline solution. As expected, the administration of CCl$_4$ led to a significant increase in both hepatic damage biomarkers in all relevant groups. Only treatment with *C. comatus* at the highest dose hindered CCl$_4$-induced elevation of aminotransferases to a certain extent.

Further, we measured MDA concentration as an indicator of lipid peroxidation, a key event in liver cell destruction caused by oxidative stress. MDA is an aldehyde which is generated from polyunsaturated fatty acids of cell membranes in a series of chain reactions. MDA reacts with thiobarbituric acid, forming a red-coloured product, the intensity of which can be measured with a spectrophotometer [8,14].

Compared with the control groups (Sal and C3), the lipid peroxidation intensity of the CCl$_4$-treated groups was significantly increased ($p < 0.001$). Administration of *C. comatus* inhibited CCl$_4$-induced lipid peroxidation in a dose-dependent manner. In the study of Stojković *et al.* [12], the lipid peroxidation inhibition ability of *C. comatus* was measured by an *in vitro* method. Interestingly, their test showed far more protective results for cultivated *C. comatus* than for wild-growing. Many living organisms have developed effective defences against oxidative stress, including preventive mechanisms, repair mechanisms and antioxidant defences. Antioxidant defences can be enzymatic and non-enzymatic, and mushrooms are thought to use both antioxidant systems. Glutathione, as the primary representative of non-enzymatic antioxidants, was assessed in this study, as well as two members of the enzymatic group, glutathione peroxidase (GPx) and catalase (CAT) (table 5). Glutathione has a thiol group, which has reducing properties and plays a crucial role in detoxifying the reactive metabolites of CCl$_4$ [8,10]. This may explain the lower concentration of GSH observed in the group of animals treated with CCl$_4$ (group SalCCl$_4$) compared with the group treated with saline only. This finding is consistent with the results of other authors, suggesting that GSH was used in reactions with toxic metabolites of CCl$_4$ [43]. By contrast, the 42-day treatment with *C. comatus* in the highest dose of 3.34 g kg$^{-1}$ increased the level of GSH more than three times in comparison with the saline only group (2.247 : 0.654; $p < 0.001$). Long-term treatment with *C. comatus* (in all three doses) before the administration of CCl$_4$ helped to maintain the GSH level (relative to control), probably by creating significant reserves of that endogenous antioxidant.

**Table 5.** Effect of long-term *C. comatus* administration on liver enzymes (aspartate and alanine aminotransferase) and various parameters of oxidative stress in rats with CCl$_4$-induced liver injury (mean ± s.e.m.).

| groups | AST | ALT | LPx | GSH | GPx | CAT |
|---|---|---|---|---|---|---|
| Sal | 118.40 ± 19.40[b] | 34.40 ± 2.62[b,c] | 0.324 ± 0.034[b] | 0.654 ± 0.028[b,c] | 1.43 ± 0.05[b] | 28.29 ± 2.01 |
| SalCCl$_4$ | 1963.00 ± 166.41 | 1110.75 ± 141.03 | 0.990 ± 0.088 | 0.395 ± 0.030 | 0.76 ± 0.01 | 25.91 ± 1.07 |
| C1CCl$_4$ | 1588.25 ± 154.34[a,c] | 802.75 ± 134.27[a,c] | 0.755 ± 0.060[a,b,c] | 0.698 ± 0.045[c] | 0.73 ± 0.04[a,c] | 26.39 ± 2.65 |
| C2CCl$_4$ | 1475.50 ± 105.77[a,c] | 870.75 ± 94.41[a,c] | 0.742 ± 0.030[a,b,c] | 0.695 ± 0.027[b,c] | 0.91 ± 0.04[a,c] | 29.47 ± 1.79 |
| C3CCl$_4$ | 1163.20 ± 112.31[a,b,c] | 567.80 ± 90.31[a,b,c] | 0.688 ± 0.048[a,b,c] | 0.828 ± 0.045[b,c] | 0.90 ± 0.03[a,c] | 27.52 ± 1.49 |
| C3 | 116.00 ± 13.52[b] | 15.00 ± 3.04[b] | 0.340 ± 0.032[b] | 2.247 ± 0.064[b] | 1.48 ± 0.07[b] | 28.01 ± 1.80 |

[a]$p < 0.05$ versus Sal group.
[b]$p < 0.05$ versus SalCCl$_4$ group.
[c]$p < 0.05$ versus C3 group.

GPx, an enzyme containing selenium, plays a primary role in the breakdown of $H_2O_2$ with the help of glutathione. Besides GPx, CAT takes a significant part in minimizing oxidative damage. CAT is also involved in the reduction of hydrogen peroxides and hydroxyl radicals. Substances that can induce the activity of GPx and CAT are proven to be potent antioxidants [8,10,43]. Activities of GPx and CAT in the liver of animals from this study are presented in table 5. There were no significant differences among groups concerning CAT activity. The administration of *C. comatus* in a dose of 3.34 g kg$^{-1}$ for 42 days did not change the activity of GPx in comparison with saline only treatment. All CCl$_4$ groups of animals had significantly lower GPx activity than the previous two groups. However, there was a slight increase of this enzyme's activity in animals that had received *C. comatus* prior to CCl$_4$ (in doses of 1.67 and 3.34 g kg$^{-1}$), but without statistical significance. Yu *et al*. [10] stated that selenium-polysaccharides and free polysaccharides from the mycelia of *C. comatus* possess antioxidant properties. They measured GPx and CAT activity in the liver and kidneys of diabetic animals and showed that treatment with selenium-polysaccharides produced more potent antioxidant activity. As stated above, the examined preparation of cultivated *C. comatus* is a good source of selenium.

A wide range of compounds from mushrooms, such as phenolics and polysaccharides, may be responsible for the observed *in vivo* antioxidant effect. Among these, *C. comatus* contains a significant quantity of tocopherols and carotenoids. Phenolic compounds exhibit antioxidant activity in biological systems, acting as free radical inhibitors, peroxide decomposers, metal inactivators or oxygen scavengers, the latter being the best described [47]. However, a recent study that determined the total antioxidant activity of the selected extracts of the mushroom *C. comatus* using a polarographic hydroxoperhydroxomercury (II) complex (HPMC) formation assay found that quinic acid exhibited the most promising antioxidant potential. This compound belongs to the class of cyclic polyols, an under-appreciated class of organic compounds, compared with aromatic (poly) phenolics [48]. Two main polysaccharide components (Ccp-I-A, Ccp-I-B) from the fruit bodies of *C. comatus* also have a strong antioxidant effect, whose scavenging activity on superoxide anions could reach 95% that of ascorbic acid [49]. Alkalic-extractable polysaccharides showed strong antioxidant and anti-inflammatory abilities as well as low hepatic lipid peroxidation levels in a model of alcohol-induced liver injury [50]. Anthraquinones, natural food additives that can be used in oil processing, meat processing, fruit and vegetable preservation and beverage production, also have extensive and important pharmacological effects, including antioxidant properties. A combination of different edible mushrooms, such as *A. bisporus*, *C. comatus* and *H. erinaceus*, promote the production of anthraquinone when fermented together [51]. Regular intake of *C. comatus,* rich in components with nutraceutical properties, can constitute an important component of diet with beneficial effects for health promotion and prevention.

### 3.2.2. Histopathological morphometric and immunohistochemical analysis

Histological analysis of standard H&E-stained liver tissue in control groups Sal and C3 showed a fully preserved cytoarchitecture (figure 2*a,f*). Hepatocytes were properly arranged in the form of Remak trabeculae, among which were the sinusoidal spaces of the usual width. Portal venule spaces were correctly positioned without inflammatory elements.

The administration of carbon tetrachloride alone in the SalCCl$_4$ group produced diffuse hepatocyte damage seen as centrilobular necrosis. Most acidophilic, necrotic altered hepatocytes were localized around the central vein, while from the centre to the lobule periphery, ballooning degeneration with a predominance of fat degenerated hepatocytes were detected (figure 2*b*). Changes were detected in almost all hepatocytes, at minimum in the form of micro/macrovesicular steatosis, while a small percentage of hepatocytes remained preserved. Rarely, hepatocytes were affected by the apoptotic process, while an inflammatory response was very mild, almost absent. Pretreatment with *C. comatus*, in three different concentrations, resulted in a particular protective effect (figure 2*c,d*,e). Although the use of morphometric analysis in C1CCl$_4$ and C2CCl$_4$ groups showed significant tissue damage, a statistically significant decrease in the surface density of necrotic hepatocytes compared with the SalCCl$_4$ group was detected (table 6).

Differences were not detected regarding the occurrence of ballooning degeneration. However, the number of fatty degenerated hepatocytes decreased significantly in the C1CCl$_4$ group and especially in C2CCl$_4$, compared with the SalCCl$_4$ group. In animals treated with *C. comatus*, more than a third of preserved hepatocytes were detected after CCl$_4$ administration, which was not observed in the control group. The highest dose of mushroom suspension administered in the C3CCl$_4$ group showed the most prominent hepatoprotective effect (figure 2*e*). The reduction in surface density of necrotic hepatocytes was statistically significant compared with the SalCCl$_4$ group but also compared with groups treated with lower mushroom doses, C1CCl$_4$ and C2CCl$_4$ (table 6).

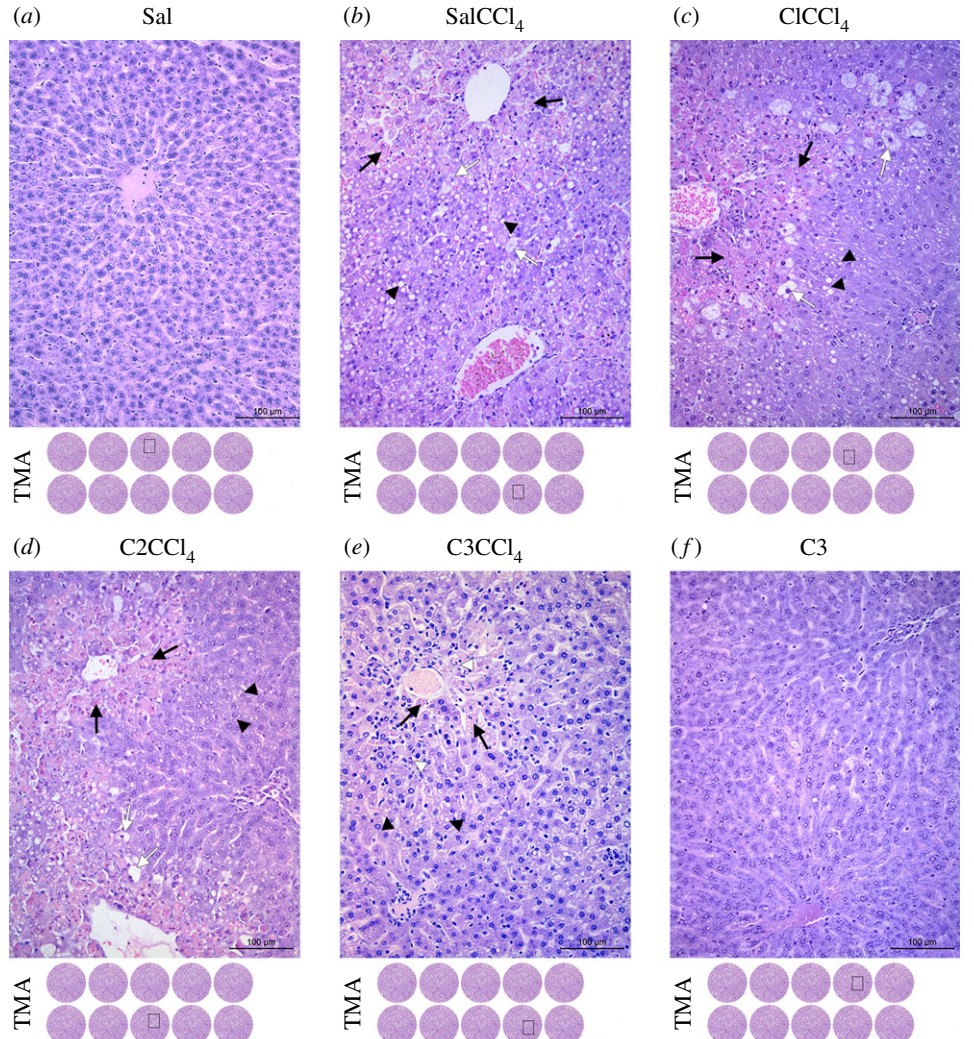

**Figure 2.** Histopathological effects of *C. comatus* on CCl$_4$-induced liver injury. Necrosis (black arrow); hepatocellular ballooning (white arrow); micro/macrovesicular steatosis (black arrowhead); inflammatory cells (white arrowhead); haematoxylin and eosin; magnification 200 x, scale bar 100 μm.

**Table 6.** Observed histopathological effects of long-term *C. comatus* administration on CCl$_4$-induced liver injury in rats (mean ± s.e.m.). Results of histopathological changes are expressed in relative percentage.

| groups | necrosis | hepatocellular ballooning | micro/macro vesicular steatosis | apoptotic hepatocytes | inflammatory cells/activity | preserved hepatocytes |
|---|---|---|---|---|---|---|
| Sal | 0 ± 0 | 0 ± 0 | 0 ± 0 | 0 ± 0 | 0 ± 0 | 100 ± 0 |
| SalCCl$_4$ | 46.12 ± 1.60 | 3.55 ± 0.69 | 45.21 ± 1.40 | 1.65 ± 0.16 | 0.94 ± 0.13 | 2.53 ± 1.03 |
| C1CCl$_4$ | 24.14 ± 2.61[a,b] | 4.98 ± 1.44[b] | 33.70 ± 5.76 | 2.16 ± 0.29[b] | 1.98 ± 0.55[b] | 33.04 ± 5.93[a] |
| C2CCl$_4$ | 23.65 ± 3.03[a,b] | 4.29 ± 1.24 | 26.49 ± 3.53[a,b] | 1.72 ± 0.18 | 2.86 ± 0.33[a,b] | 40.99 ± 3.76[a] |
| C3CCl$_4$ | 10.15 ± 0.65[a] | 0.82 ± 0.21 | 40.52 ± 0.74 | 1.28 ± 0.16 | 5.33 ± 0.59[a] | 41.90 ± 0.57[a] |
| C3 | 0 ± 0 | 0 ± 0 | 0 ± 0 | 0 ± 0 | 0 ± 0 | 100 ± 0 |

[a]$p < 0.05$ versus SalCCl$_4$ group.

[b]$p < 0.05$ versus C3CCl$_4$ group,

Results of Sal and C3 group were excluded from statistical analysis because none of the histopathological changes were observed in animals of these groups.

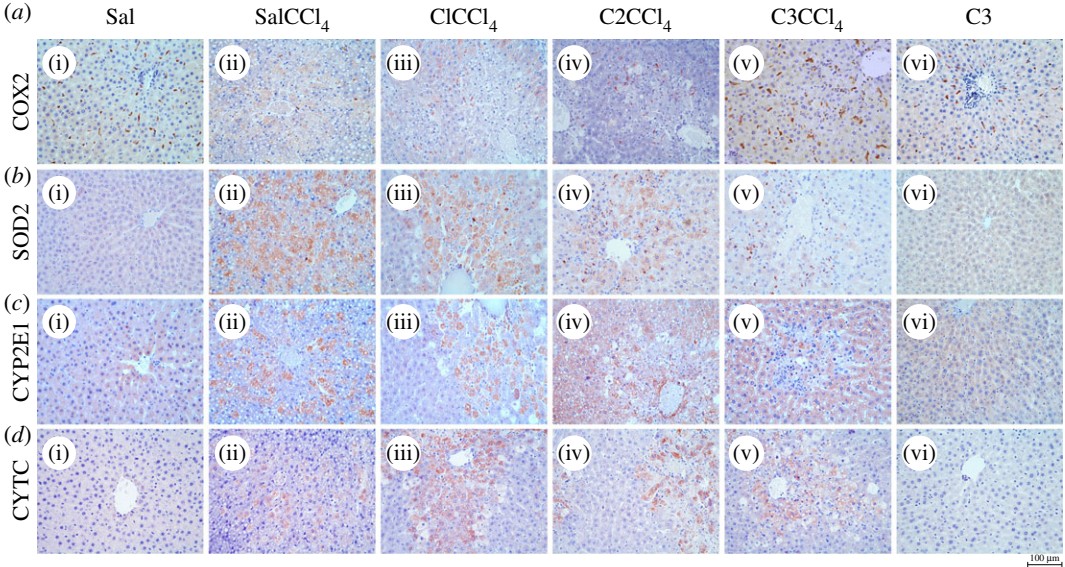

**Figure 3.** Immunohistochemical analysis of effects of *C. comatus* on CCl$_4$-induced liver injury. COX2 *a*(i–vi); SOD2 *b*(i–vi); CYP2E1 *c*(i–vi); CYTC *d*(i–vi); magnification 200 x; scale bar 100 μm.

Also, ballooning hepatocyte degeneration was practically not observed, while on the other hand, the number of hepatocytes with fatty degeneration increased in comparison with groups treated with lower mushroom doses. The increase in steatosis was inversely proportional to the number of necrotic hepatocytes, which leads to the conclusion that damage to the liver tissue was more reversible if animals were pretreated with a higher dose of mushroom. The number of apoptotic hepatocytes did not differ between the experimental groups, but the intensity of the inflammatory reaction increased significantly in animals treated with higher doses of mushrooms (table 6). Histological findings from our previous study [14] and the study of Jayakumar *et al.* [43] gave us an idea that treatment with mushrooms can oppose oxidative stress. We have performed a morphometric analysis of liver tissue to generate results that could be processed statistically [25]. To the best of our knowledge, this is the first report on morphometric analysis of liver tissue of animals treated with mushrooms. Results from this analysis suggest that *C. comatus* possesses hepatoprotective effects, and this is in concordance with previously mentioned papers [8,43].

For a better understanding of *C. comatus* action, we did an immunohistochemical analysis. The immunohistochemical marker COX2 shows the dominant positivity in Kupffer cells (KCs) of the liver. When analysing the groups Sal and C3, the same presentation of KCs located inside the sinusoidal capillaries was observed (figure 3*a*i,vi). The administration of CCl$_4$ in the SalCCl$_4$ group reduced to a great extent the number of KCs, while poor COX2 immunopositivity was also observed in necrotic hepatocytes localized around the central vein (figure 3*a*ii). A slightly higher number of KCs was observed in groups C1CCl$_4$ and C2CCl$_4$ (figure 3*a*iii,iv) as well as the identical staining of damaged hepatocytes. In animals pretreated with the highest dose of mushroom before CCl$_4$ administration (group C3CCl$_4$), the number of KCs was significantly increased (figure 3*a*v). The use of SOD2 immunohistochemical oxidative stress marker in Sal and C3 (figure 3*b*i,vi) groups did not result in the staining of liver structures. However, the use of CCl$_4$ (figure 3*b*ii,iii,iv,v) induced the expression of this marker in the cytoplasm of altered hepatocytes. Also, SOD2 positivity could be observed in neutrophilic granulocytes, whose number increased in a *C. comatus* dose-dependent manner. Thus, the smallest number of SOD2 positive hepatocytes with the highest number of positive neutrophilic granulocytes was found in the C3CCl$_4$ group.

The immunohistochemical markers COX2 and SOD2 indicated that the administration of mushroom *C. comatus* induces reparation processes in the liver, as manifested through increased macrophage activity of COX2 positive KCs, as well as granulocyte activity of SOD2 positive neutrophilic granulocytes. On the other hand, by reducing the number of necrotic hepatocytes with the administration of higher mushroom doses, the signs of oxidative stress within the hepatocytes (SOD2 positivity) were also reduced. As stated in the study of Jayakumar *et al.* [43], KCs indeed release quantities of reactive oxygen species but also other bioactive agents. These endogenous bioactive substances assist in the clearing of irreversibly damaged hepatocytes and limiting further devastation caused by CCl$_4$ products that are formed within the liver cells [43].

Immunohistochemical staining of CYP2E1 in Sal and C3 groups resulted in a weak diffuse cytoplasmic positivity of the hepatocytes located around the central vein. By contrast, necrotic and fat degenerated hepatocytes in CCl4-treated groups (figure 3cii,iii,iv,v) were characterized by intense cytoplasmic staining. In C2CCl4 and C3CCl4 groups, a surprising, pronounced cytoplasmic staining for CYP2E1 was observed in surrounding, preserved hepatocytes, suggesting an increase in proapoptotic P450 enzyme activity in these healthy hepatocytes. Immunohistochemical analysis in Sal and C3 groups resulted in a complete lack of positivity for the CYTC marker of apoptosis (figure 3di, vi). The administration of CCl4 resulted in positivity for this proapoptotic marker. Namely, in the SalCCl4 group the positive reaction is slightly weaker, and it is present in individual necrotic and fat-altered hepatocytes around the central vein (figure 3dii). On the other hand, pretreatment with *C. comatus* in groups C1CCl4, C2CCl4 and C3CCl4 resulted in somewhat more intense positivity of necrotic and fat-degenerated hepatocytes (figure 3diii,iv,v).

Taking all immunohistochemical results into account, it seems that the hepatoprotective effects of *C. comatus* in oxidative stress-induced liver damage involve the induction of reparatory mechanisms. Long-term administration of *C. comatus* appeared to increase the antioxidant resources available to the liver challenged by CCl4, thus limiting its damaging effects. The main limitation of the present, and other similar, studies using the CCl4 model of oxidative stress is the fact that according to the protocol, sacrificing of animals must occur one day after the administration of CCl4 [8,36,43]. A longer follow-up period would allow for estimation of the completeness of liver reparation and restoration processes required to make definitive conclusions about the hepatoprotective and reparatory potential of *C. comatus*.

# 4. Conclusion

Findings from this study imply that cultivated *C. comatus* is an excellent dietary source of nutritional and bioactive compounds. Rich in nutrients, with low toxic metal content, it represents a safe addition to an everyday diet. Furthermore, results also imply that cultivated *C. comatus* could be an excellent source of antioxidant compounds that could help to improve the defence system against free radical induced damage. Thus, the present *in vivo* study of the long-term administration of cultivated *C. comatus* supports its antioxidant effects, described in our previously published papers. Because of complex composition, more investigations are needed to demonstrate whether the beneficial activities are accomplished by one active compound, or combined activities of different compounds that exist in the extract. Furthermore, future studies are needed to decipher the exact molecular mechanisms responsible for the observed beneficial effects of *C. comatus*. Still, *C. comatus* can serve as an easily accessible food, rich in natural antioxidants, as well as an additive or ingredient for use in the formulation of nutraceuticals and functional foods.

Ethics. The experimental procedures were conducted under the European Directive (2010/63/EU) for animal experiments, and they were reviewed and approved by the Ethics Committee for Protection and Welfare of Experimental Animals at the University of Novi Sad, Serbia (Approval No. III-2011-07).
Data accessibility. Articles supporting the data, and digital research materials are available on the University of Novi Sad repository https://oblak.uns.ac.rs/index.php/s/fzT4Kxqjj52By4q.
Authors' contributions. N.S. and S.V. performed the animal experiments and analysis; I.Č. performed the histological examination and drafted the article. A.T. analysed the data and drafted the manuscript. A.R., M.P. and A.S. designed the study and were involved in article revision after drafting. All authors approved the current version and take full responsibility for the information contained within the manuscript.
Competing interests. We declare we have no competing interests.
Funding. This work was supported by the Ministry of Education, Science and Technological Development of the Republic of Serbia (grant nos. 41012 and 172050).
Acknowledgement. We thank Gaewyn Ellison, PhD from School of Pharmacy and Biomedical Sciences, Curtin Health Innovation Research Institute, Perth, Australia, for improving the use of English in the manuscript.

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
