## [Reviewer comments · Royal Society Open Science]

Review History

RSOS-200900.R0 (Original submission)

Review form: Reviewer 1

Is the manuscript scientifically sound in its present form?

Yes

Are the interpretations and conclusions justified by the results?

Yes

Is the language acceptable?

Yes

Do you have any ethical concerns with this paper?

No

Have you any concerns about statistical analyses in this paper?

No

Recommendation?

Major revision is needed (please make suggestions in comments)

Comments to the Author(s)

Major issues:

1, relevant information on the animals used in the study should be provided. For example, numbers of females and males in each group, body weight changes during the experiment, healthy status of the rats. These are important information for the readers to understand the safety of the mushroom extracts.

2, To make the study more conclusive, the authors should conduct a western blot assay to check the biomarker protein levels in the liver. This should be done consistent with the immunochemistry assay.

3, the English of the manuscript must be refined.

4, a graphical figure showing the workflow and study design should be added to the manuscript. It would help the readers to understand the main idea of the study.

5, the authors should discuss in the manuscript the potential mechanisms of the anti-oxidant effect of the mushroom extracts. Is it the effect of flavonoids? What are the possibilities. What is the future direction? Relevant topics should be included and explored in the manuscript.

Review form: Reviewer 2 (Ahmed E. Abdel Moneim)

Is the manuscript scientifically sound in its present form?

Yes

Are the interpretations and conclusions justified by the results?

Yes

Is the language acceptable?

No

Do you have any ethical concerns with this paper?

No

Have you any concerns about statistical analyses in this paper?

No

Recommendation?

Major revision is needed (please make suggestions in comments)

Comments to the Author(s)

- In the abstract " that not only wild growing but also cultivated *C. comatus* could be a good source of nutritional and bioactive compounds." Are the authors included wild-growing mushrooms to say this?

- There are many English errors. The work should be revised by an English native speaker.

- After mentioned the full name of *Coprinus comatus*, write *C. comatus*.

- Some references are old references and there are new versions of the manuscript. Please use the new papers instead of the old papers.

- It is hard to understand the statistical analysis results.

Decision letter (RSOS-200900.R0)

Dear Dr Tomas:

Title: Chemical composition, nutritional profile and in vivo antioxidant properties of the cultivated mushroom *Coprinus comatus*
Manuscript ID: RSOS-200900

The editor assigned to your manuscript has now received comments from reviewers. We would like you to revise your paper in accordance with the referee and Subject Editor suggestions which can be found below (not including confidential reports to the Editor). Please note this decision does not guarantee eventual acceptance.

Please submit your revised paper before 18-Jul-2020. Please note that the revision deadline will expire at 00.00am on this date. If we do not hear from you within this time then it will be assumed that the paper has been withdrawn. In exceptional circumstances, extensions may be possible if agreed with the Editorial Office in advance. We do not allow multiple rounds of revision so we urge you to make every effort to fully address all of the comments at this stage. If deemed necessary by the Editors, your manuscript will be sent back to one or more of the original reviewers for assessment. If the original reviewers are not available we may invite new reviewers.

RSC Associate Editor:
Comments to the Author:
(There are no comments.)

RSC Subject Editor:
Comments to the Author:
(There are no comments.)

Reviewers' Comments to Author:
Reviewer: 1

Comments to the Author(s)
major issues:

1, relevant information on the animals used in the study should be provided. For example, numbers of females and males in each group, body weight changes during the experiment, healthy status of the rats. These are important information for the readers to understand the safety of the mushroom extracts.

2, To make the study more conclusive, the authors should conduct a western blot assay to check the biomarker protein levels in the liver. This should be done consistent with the immunochemistry assay.

3, the English of the manuscript must be refined.

4, a graphical figure showing the workflow and study design should be added to the manuscript. It would help the readers to understand the main idea of the study.

5, the authors should discuss in the manuscript the potential mechanisms of the anti-oxidant effect of the mushroom extracts. Is it the effect of flavonoids? What are the possibilities. What is the future direction? Relevant topics should be included and explored in the manuscript.

Reviewer: 2

Comments to the Author(s)

- In the abstract " that not only wild growing but also cultivated *C. comatus* could be a good source of nutritional and bioactive compounds." Are the authors included wild-growing mushrooms to say this?
- There are many English errors. The work should be revised by an English native speaker.
- After mentioned the full name of *Coprinus comatus*, write *C. comatus*.
- Some references are old references and there are new versions of the manuscript. Please use the new papers instead of the old papers.
- It is hard to understand the statistical analysis results.

Author's Response to Decision Letter for (RSOS-200900.R0)

See Appendix A.

RSOS-200900.R1 (Revision)

Review form: Reviewer 1

Is the manuscript scientifically sound in its present form?

Yes

Are the interpretations and conclusions justified by the results?

Yes

Is the language acceptable?

Yes

Do you have any ethical concerns with this paper?

No

Have you any concerns about statistical analyses in this paper?

No

Recommendation?

Accept as is

Comments to the Author(s)

The authors have addressed my concerns. I suggest to accept the manuscript.

Decision letter (RSOS-200900.R1)

Dear Dr Tomas:

Title: Chemical composition, nutritional profile and in vivo antioxidant properties of the cultivated mushroom *Coprinus comatus*

Manuscript ID: RSOS-200900.R1

It is a pleasure to accept your manuscript in its current form for publication in Royal Society Open Science. The chemistry content of Royal Society Open Science is published in collaboration with the Royal Society of Chemistry.

RSC Associate Editor:
Comments to the Author:
(There are no comments.)

RSC Subject Editor:
Comments to the Author:
(There are no comments.)

Reviewer(s)' Comments to Author:
Reviewer: 1

Comments to the Author(s)
The authors have addressed my concerns. I suggest to accept the manuscript.

Appendix A

Dear Editors,

*Our team is pleased to resubmit the revised version of the manuscript, entitled "Chemical composition, nutritional profile and in vivo antioxidant properties of the cultivated mushroom *Coprinus comatus*".*

We appreciated the constructive criticisms of the reviewers. Comments of the reviewers were highly insightful and enabled us to greatly improve the quality of our manuscript. We have addressed each of their concerns, as outlined below. In this letter, we tried to state point-by-point the changes we have made to the article.

REVIEWER 1

1. Relevant information on the animals used in the study should be provided. For example, numbers of females and males in each group, body weight changes during the experiment, healthy status of the rats. These are important information for the readers to understand the safety of the mushroom extracts.

The method section is now expanded and rewritten according to the suggestions of the reviewer. Information about number of animals of different sex, as well as information about monitoring of health status during the experiment, is now included in the method section.

Also, we have added an additional table presenting changes in their body weight during the experiment, as this offers the important information about the safety of examined extract.

2. To make the study more conclusive, the authors should conduct a western blot assay to check the biomarker protein levels in the liver. This should be done consistent with the immunochemistry assay.

*We appreciate the suggestion to conduct a western blot assay, but the current situation and time constraints make it difficult to procure resources necessary to carry out the experiments mentioned. We are aware that WB is often used together with IH as a complementary assay to confirm antibody specificity and provide more quantitative analysis of protein levels, but we believe that the performed experiments offer strong evidence about beneficial effects of the long-term administration of *C. comatus* in vivo, which is scarcely reported in the literature. We have observed consistent results when determining levels of biochemical parameters indicative of liver injury and markers of oxidative stress. The histological and immunohistochemical study provided further insight into the effects of *C. comatus*, showing that the observed changes in levels of mentioned markers are in agreement to histological findings. In the future, we will try to include the protein biomarker quantification to better explain the mechanisms responsible for the observed effects.*

3. The English of the manuscript must be refined.

The whole paper has been thoroughly reviewed and edited by the native English speaker with a strong academic background who has provided a signed statement that the language editing was performed.

4. A graphical figure showing the workflow and study design should be added to the manuscript. It would help the readers to understand the main idea of the study.

In methods section, a graphical figure showing the workflow has been added (Supp. File 1)

5. The authors should discuss in the manuscript the potential mechanisms of the anti-oxidant effect of the mushroom extracts. Is it the effect of flavonoids? What are the possibilities? What is the future direction? Relevant topics should be included and explored in the manuscript.

Potential mechanisms of antioxidant effects of the mushroom extracts have been included in the discussion section with special focus on the substances present in mushrooms which might be responsible for the observed, marked antioxidant effects. The conclusion section is now broadened and includes the future direction of the following studies.

REVIEWER 2

- In the abstract "that not only wild growing but also cultivated *C. comatus* could be a good source of nutritional and bioactive compounds." Are the authors included wild-growing mushrooms to say this?

The sentence has been removed from the abstract as it did not add to the clarity.

- There are many English errors. The work should be revised by an English native speaker.

The whole paper has been thoroughly reviewed and edited by the native English speaker with a strong academic background who has provided a signed statement that the language editing was performed.

- After mentioned the full name of *Coprinus comatus*, write *C. comatus*.

The change suggested has been made.

- Some references are old references and there are new versions of the manuscript. Please use the new papers instead of the old papers.

The reference list has been updated to include the latest versions of the manuscripts cited. We have also updated the manuscript with a number of new, recent references relevant to the topic.

- It is hard to understand the statistical analysis results.

We have modified the way statistical significance was presented in the tables, which we believe now makes the results easier to comprehend.